# Experimental and Modeling Study of the Fabrication of Mg Nano-Sculpted Films by Magnetron Sputtering Combined with Glancing Angle Deposition

**Hui Liang [1,2], Xi Geng [1], Wenjiang Li [1,3,*], Adriano Panepinto [2], Damien Thiry [2], Minfang Chen [1,3,*] and Rony Snyders [2,4,*]**

[1] School of Materials Science and Engineering, Tianjin University of Technology, Tianjin 300384, China; hui.liang@umons.ac.be (H.L.); xigeng0220@126.com (X.G.)

[2] Chimie des Interactions Plasma-Surface, University of Mons, 20 Place du Parc, B 7000 Mons, Belgium; adriano.panepinto@umons.ac.be (A.P.); Damien.THIRY@umons.ac.be (D.T.)

[3] Key Laboratory of Display Materials and Photoelectric Device (Ministry of Education), Tianjin 300384, China

[4] Materia Nova Research Center, 1 Avenue Nicolas Copernic, B 7000 Mons, Belgium

\* Correspondence: liwj@tjut.edu.cn (W.L.); jgj@tjut.edu.cn (M.C.); rony.snyders@umons.ac.be (R.S.); Tel.: +32-65-554955 (R.S.)

**Abstract:** Today, Mg is foreseen as one of the most promising materials for hydrogen storage when prepared as nano-objects. In this context, we have studied the fabrication of Mg nano-sculpted thin films by magnetron sputtering deposition in glancing angle configuration. It is demonstrated that the microstructure of the material is controllable by tuning important deposition parameters such as the tilt angle or the deposition pressure which both strongly affect the shadowing effect during deposition. As an example, the angle formed by the column and the substrate and the intercolumnar space varies between ~20° to ~50° and ~45 to ~120 nm, respectively, when increasing the tilt angle from 60° to 90°. These observations are highlighted by modeling the growth of the material using kinetic Monte Carlo methods which highlights the role of surface diffusion during the synthesis of the coating. This work is a first step towards the development of an air-stable material for hydrogen storage.

**Keywords:** Mg columnar films; glancing angle deposition; magnetron sputtering; kinetic Monte Carlo modeling

## 1. Introduction

With increased worldwide energy consumption that is associated with the global warming problem and the depletion of fossil fuels, renewable energy sources from hydro, solar, and wind sources are increasingly replacing the conventional fuels [1,2]. This is the driving force of a real appeal for the development of new solutions in several domains of our society, including the transport industry. Considering the latter, today, several strategies are considered to design the car of the future, and among them, the hydrogen car is one of the most promising ones. Indeed, hydrogen can be produced by various electrochemical and biological methods and has a higher chemical energy as compared with fossil fuels [3–5]. Furthermore, once produced from any energy source, hydrogen generates electricity during fuel cell operations, leaving water vapor as the only exhaust gas, without any other greenhouse gases or harmful emissions [2]. Nevertheless, several issues related to the production, distribution, and storage of hydrogen have to be fixed before using hydrogen as an economically viable fuel for the transport industry [6]. In particular, the hydrogen storage is an important issue related to the low volumetric density of hydrogen. Among the solutions developed to store hydrogen, the utilization of solid-state materials is preferred because of its higher volumetric density (as compared with gaseous

and liquid solutions) and for safety reasons. Among the solid-state materials that store hydrogen, the hydride materials where hydrogen is chemically bounded (i.e., not only adsorbed) appear to be good candidates [7–9].

Specifically, magnesium-based hydrides, and more specifically elemental magnesium hydride ($MgH_2$), are often considered as promising materials for hydrogen storage because magnesium (Mg) is abundant, low cost, has low density, low toxicity and higher hydrogen capacity and reversibility as compared with other hydrides [10,11]. Nevertheless, this material suffers two main drawbacks which are a high desorption temperature and a slow hydrogen sorption kinetic [11]. In addition, Mg can easily be oxidized by oxygen and hydrogen not easily diffused in bulk Mg.

For many years, these problems have been addressed by the community. A complete review on the topic has recently been published by Sadhasivam et al. [12]. From these works, it appears that the reduction of the size of the Mg compounds down to the nanoscale strongly improves the thermodynamic properties of the material [13]. Therefore, several routes have been investigated to reduce the size of the Mg/$MgH_2$ (below 1 µm) particles such as mechanical ball milling in the presence (or not) of catalyst materials leading to significant improvements in term of the sorption kinetic of the material [14]. Nevertheless, if the sorption kinetic is improved by this approach, this is not the case for the thermodynamic parameters [14]. In order to overcome this problem, it has been suggested that a further reduction of the dimension (<100 nm) of the material could help. This is why efforts have been developed in order to fabricate 1-, 2- or 3D Mg nanoparticles [6,10]. As an example, Barawi et al. [15,16] reported on the synthesis of Mg films by e-beam evaporation on $SiO_2$ substrates with a thickness ranging from 45 to 900 nm and demonstrated that it plays a major role in the hydrogen absorption kinetics.

In this context and in view of the material science challenges, plasma techniques appear as an ideal technological platform to synthesize these materials. Indeed, these technologies are known as "green" technologies, since they allow for good control of the material properties and their industrial transfer has been demonstrated in many fields such as the glass industry or microelectronics [17–21]. Among these technologies, magnetron sputtering (MS) is usually used to grow dense thin films of various materials (from metal to polymer coatings) [21–23]. Nevertheless, when used in the glancing angle deposition (GLAD) configuration, it has been demonstrated that nanostructurated coatings for the microstructure can be controlled. As an example, we recently reported on the growth of Ti and $TiO_2$ nanostructurated films by using this approach [19,20].

Therefore, in this work, we aim to study the growth of nano-sculpted Mg films by combining magnetron sputtering and glancing angle deposition (MSGLAD) in order to better understand the growth mechanism of this material which could ultimately be used in composite material for hydrogen storage application. Our strategy consists of a systematic study of the influence of important deposition parameters namely the tilt angle ($\alpha$) and the working pressure ($P_{Tot}$) on the microstructure of the synthesized material. These experimental results are compared to computer simulation by Kinetic Monte Carlo (KMC) using the NASCAM code to better understand the growth mechanism of the Mg thin films.

## 2. Material and Method

### 2.1. Experimental

All experiments were carried out in a cylindrical stainless-steel chamber (height: 60 cm, diameter: 42 cm), shown in Figure 1. The chamber was evacuated by a turbo-molecular pump (Edwards nEXT400D 160W, Burgess Hill, UK), down to a residual pressure of $10^{-7}$ Pa. A magnetron cathode was installed at the top of the chamber and the substrate was located at a distance of 80 mm. A 2-inch in diameter and 0.25-inch thick Mg target (99.99% purity) was used. The target was sputtered in DC mode using an Advanced Energy MDK 1.5 K power supply in argon atmosphere using a flow of

12 sccm. Conductive silicon wafers (100) were used as substrates and rinsed with ultra-pure water before deposition.

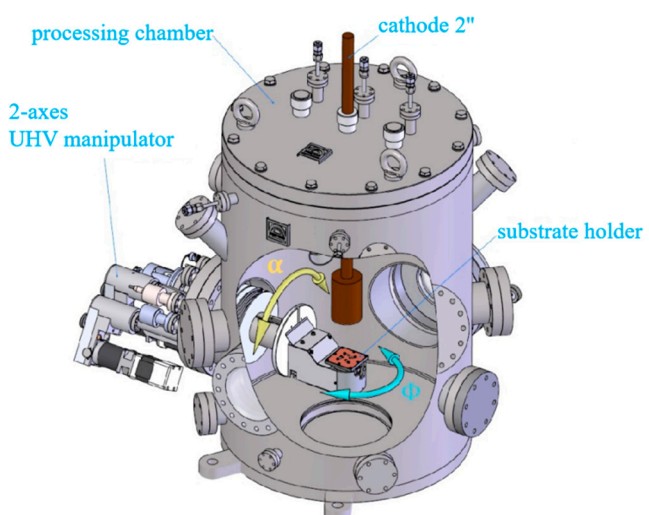

**Figure 1.** Sketch of the deposition chamber used in this work.

Using the GLAD system, the substrate can be tilted with an angle α and eventually rotated by an angle (φ) either step-by-step or with a continuous angular speed in order to generate diverse thin film architectures. In this work, we have only studied the effect of the tilt angle on the architecture of the deposited films with α = 60°, 80°, 82.5°, 85°, 87°, and 89°. On the other hand, we also evaluated the influence of the working pressure ($P_{Tot}$) which was varied from 0.13 to 1.3 Pa. For all deposition, the sputtering power was kept constant at 50 W and the deposition time varied between 10 and 20 min depending on the deposition conditions in order to reach similar thicknesses for all deposited films.

The morphology of the material was characterized with a field emission gun scanning electron microscope (FEG-SEM, Hitachi SU8020, Ri Li, Japan). In addition, from the SEM images, we extracted the so-called aspect ratio, Γ, which is defined as the ratio between the inter-columnar space and the column width.

The chemical composition of the films was evaluated by X-ray photoelectron spectroscopy (XPS) on a VERSAPROBE PHI 5000 hemispherical analyzer from Physical Electronics with a base pressure of <3 × $10^{-7}$ Pa. The X-ray photoelectron spectra were collected at the take-off angle of 45° with respect to the electron energy analyzer, operating in constant analyzer energy (CAE) mode (23.50 eV). The spectra were recorded with the monochromatic Al Kα radiation (15 kV, 25 W) with a highly focused beam size of 100 μm. The energy resolution was 0.7 eV. Eventual surface charging was compensated for by a built-in electron gun and an argon ion neutralizer. For the chemical depth profile, an $Ar^+$ ion source was operated at 1 μA and 2 kV with a raster area of 2 mm ✕ 2 mm at an incident angle normal to the sample surface of 54.7°. The XPS spectra were referenced to the Mg2p peak at 49.5 eV arising from the metallic magnesium component [24]. Atomic compositions were derived from peak areas using photoionization cross sections calculated by Scofield, corrected for the dependence of the escape depth on the kinetic energy of the electrons and corrected for the analyzer transmission function of our spectrometer.

The thickness of the films, as measured by a mechanical profilometer Dektak 150 from Veeco, was kept constant for all films, and their average thickness was about 620 ± 20 nm. As an example, the deposition rate was 0.32 nm/s for a deposition angle of 85° and a sputtering pressure of 2 mTorr.

Finally, the phase constitution of the samples was evaluated by X-ray diffraction (XRD) using a PANalytical Empyrean diffractometer working with Cu $K_{\alpha 1}$ radiation (λ = 0.1546 nm) in the grazing incidence configuration (Ω = 0.5°). The X-ray source voltage was fixed at 45 kV and the current

at 40 mA. The grain size ($G_S$) was calculated from the XRD pattern using the following Scherrer equation [25]:

$$G_s = \frac{K\lambda}{\beta \cos\theta}$$

where, *K* is a dimensionless shape factor, λ is the X-ray wavelength, β is the diffraction line broadening at half the maximum intensity (FWHM), and θ is the Bragg angle.

*2.2. Simulation*

NASCAM is an atomistic deposition simulation code based on the kinetic Monte Carlo (kMC) method. It can be used for the modeling of different processes occurring at the surface such as the growth of films during deposition. The atoms are deposited on the substrate at random positions at an equal time interval which is determined by the deposition rate. Only diffusion or evaporation events can take place between two deposition events. Energy transfer during ballistic collision events is also taken into account. This made it suitable to simulate glancing angle deposition processes [26]. The energy and angular distribution of incident particles were calculated by SRIM [27] and SIMTRA [28]. First, SRIM was used to calculate the energy and the direction of particles which leave the target. The particles were then transported in the gas phase by the SIMTRA code which took into account all the collisions between the sputtered species and the gas molecules. The energy and the angular distribution of the species at the substrate location were derived for each working conditions by the introduction of the experimental parameters, which included the working pressure, the particles' energy which was a function of the power applied at the target, the racetrack sizes, and the target-to-substrate distance. After that, these files were used as input data for NASCAM. Other parameters could be tuned in the input file. To compare the simulation with the experience, we tuned the number of deposited atoms and the substrate size (XYZ). In these conditions, the simulated and the experimental thin film had the same thickness.

The energy and the angular distribution of the species at the substrate location were derived for each working conditions by the introduction of the experimental parameters such as the working pressure, the power applied to the target, the racetrack size, and the target-to-substrate distance, Other parameters could be varied in the NASCAM input file such as the number of deposited atoms (N) and the substrate size (XYZ). In order to compare simulated and experimental thin films, both had the same thickness. For direct comparison of the cross-sectional film morphology, "2D" (Y = 2) NASCAM simulations were performed (N = 1000 atoms), whereas, for density and porosity evaluation, "3D" (Y = 4) simulations were performed (N = 1.667 atoms). The deposition rate was fixed at 0.5 monolayer by second (0.301 nm/s), which was close to the experimental value (0.32 nm/s) at a deposition angle of 85°.

## 3. Results and Discussion

*3.1. Characterization of A Dense Mg Film*

In a first attempt, we have grown a Mg thin film in conventional geometry (α = 0°) in order to evaluate the deposition rate, as well as the chemical composition and the phase constitution of the deposited material. Figure 2 shows the survey XPS spectrum recorded for this film. It reveals the presence of Mg, O, and C lines at 49.5 eV (Mg2p), 285 eV (C1s), and 530 eV (O1s). From the quantification of these signal, ~50 atom% of oxygen and 10 atom% of carbon are observed. These are likely related to surface pollution that appears during the transport of the sample from the chamber to the XPS machine. Particularly, the presence of oxygen while working in non-reactive conditions is related strong reactivity of Mg towards $O_2$ ($\Delta H_{f(MgO)}$ = −601.8 kJ·mol$^{-1}$) [29], which allows for the oxidation of the top surface of the deposited film. In order to clarify the chemistry of the film and to validate that the presence of carbon, as well as the surface oxidation of Mg, are related to surface pollution, depth profiling of the films were performed by using an Ar$^+$ gun in the XPS machine before recording the XPS spectra during 2, 4, and 20 min of erosion. The results are presented in Table 1. Clearly, it appears from this analysis that a few minutes of erosion allows removal of all the initially observed carbon contamination

as well as reduction of the oxygen content to an ~10 atom% limit, validating the surface contamination of the as-deposited sample. The presence of the 10 atom% of oxygen in the bulk of the material is likely explained by the presence of very low quantity adsorbed water or oxygen in the deposition chamber even if the base pressure is good quality ($10^{-7}$ Torr). Indeed, because of the already mentioned strong reactivity of Mg towards $O_2$, a getter effect likely occurs and leads to the partial oxidation of the material similar to the effect already observed for another getter material such as Ti [30].

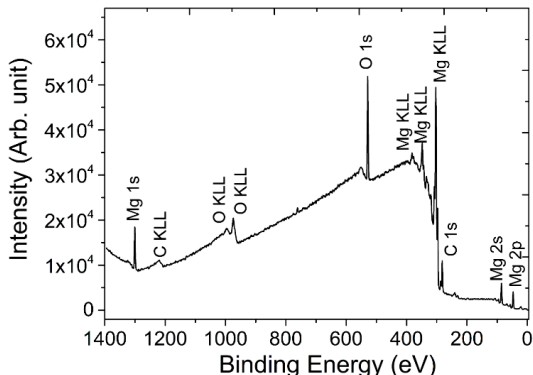

**Figure 2.** XPS survey spectra of a dense Mg film prepared for $\alpha = 0°$ and $P_{Tot} = 0.26$ Pa. The sputtering power is 50 W.

**Table 1.** Elemental composition of the as-deposited Mg thin film before and after 2 min of erosion.

| At.% Mg | | At.% O | | At.% C | |
|---|---|---|---|---|---|
| As prepared | After erosion | As prepared | After erosion | As prepared | After erosion |
| 44.6 | 89.9 | 48.9 | 10.1 | 10.5 | 0 |

In order to support this conclusion, Figure 3 shows the evolution of the Mg2p XPS line as a function of the depth profiling. The estimated sputtering rate is ~20 nm/min, according to the study reported by Milcius et al. [31]. For the as-deposited sample, it appears that the Mg line is composed of two components corresponding to metallic Mg at 49.5 eV and $Mg^{2+}$ at 50.8 eV. On the other hand, at ~60 eV, a satellite line related to the metallic component is also observable. The presence of a strong oxidized component is in line with the surface stoichiometry of the surface composition of the as-prepared sample. After two minutes of sputtering, which is evaluated to correspond to 40 nm, the oxidized component of the Mg peak completely vanishes while the satellite peak intensity strongly increases. Both these observations clearly confirm that the deposited film is only oxidized on its top surface.

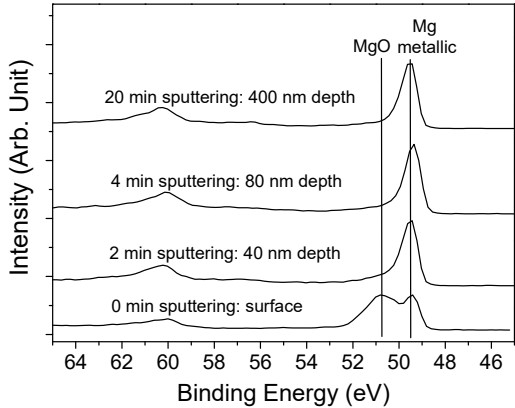

**Figure 3.** Evolution of the Mg2p line during the depth profiling of the Mg films prepared for $\alpha = 0°$ and $P_{Tot} = 0.26$ Pa.

The XRD analysis of the as-deposited dense film is presented in Figure 4. The diffractogram reveals that the film consists of a polycrystalline material with the presence of several diffraction peaks, (002), (102), and (102), attributable to the cubic phase of Mg (JCPDS card N° 04-0770) [31]. The (002) peak dominates the spectra which is likely explained by a preferential orientation of the growth along the c axis as already reported for other materials [13]. In these conditions, applying the Sherrer formula to the dominating peak, a crystal size of about 25 nm has been calculated.

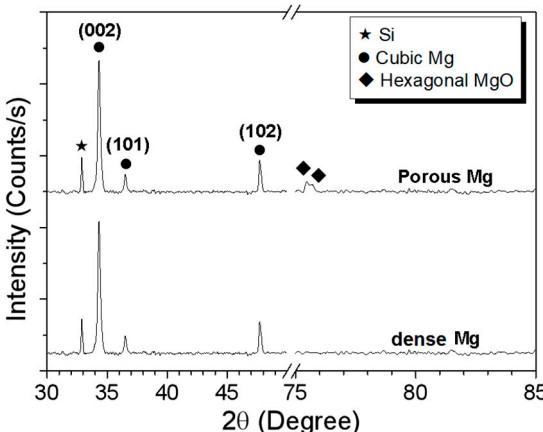

**Figure 4.** XRD spectra of the Mg films prepared for $\alpha = 0°$ and $P_{Tot} = 0.26$ Pa (dense Mg) and for $\alpha = 85°$ and $P_{Tot} = 0.26$ Pa (porous Mg). In both cases, the sputtering power is 50 W.

### 3.2. Nano-Sculpted Mg Films

After having characterized the reference film, we have studied the nanostructuration of the material by using MSGLAD by systematically studying the influence of both the tilt angle ($\alpha$) and the working pressure ($P_{Tot}$) on the morphology of Mg films evaluated by SEM images. In addition, in order to clarify and understand the growth mechanisms, the experimental data was compared to the result of the modeling of the growth by kinetic Monte Carlo method using the NASCAM software.

Figure 5 shows, as a typical example of the generated microstructure, the cross section and the surface images of a Mg thin films deposited for $\alpha = 85°$ and $P_{Tot} = 0.26$. Pa. Figure 5a shows the side view of the film which reveals that it is made of well separated faceted columns which have a width of ~200 nm, a length of ~800 nm, and (when considering the substrate surface) a tilt angle $\beta$ of ~44°. The estimated dimension of the voids between the column is ~10–60 nm. Figure 5b shows the surface of the film and confirms that the columns are well separated. Similar images for all samples synthesized by varying $\alpha$ and $P_{Tot}$ are shown in the Supplementary Figure S1. Similar morphologies have been obtained for all samples. Nevertheless, it is observed that the deposited film features ($\beta$, the column length, the intercolumnar space, and the column width) more or less depend on these parameters. This offer many knobs for tuning the morphology of the deposited films as a function of the foreseen application.

Figure 6 summaries the evolution of $\beta$ for $60° < \alpha < 89°$ (Figure 6a) and for $0.13$ Pa $< P_{Tot} < 1.3$ Pa (Figure 6b) evaluated from SEM images. Figure 6a reveals that $\beta$ strongly depends on $\alpha$ in line with previous work on Ti and TiO$_2$ nano-sculpted films. This can be explained considering the influence of the atomic shadowing effects during deposition in GLAD configuration [19]. In particular, at extremely oblique incidence angles of the flux (>60°), the shadowing mechanism is strongly enhanced and results in a porous microstructure composed of columns inclined toward the vapor source [32]. $\beta$ drastically increases for $\alpha > 60°$ and stabilizes for $\alpha > 85°$. This evolution is explained by considering that when the direction of vapor incidence is normal to the film surface, the diffusion during the accommodation of the adatoms is a few atomic distances in the isotropic direction. However, under oblique incidence, the diffusion occurs in the direction given by the projection of the vapor beam direction on the film surface. The amount of kinetic energy (momentum) preserved in the direction parallel to the film surface is only determined by the angle of incidence.

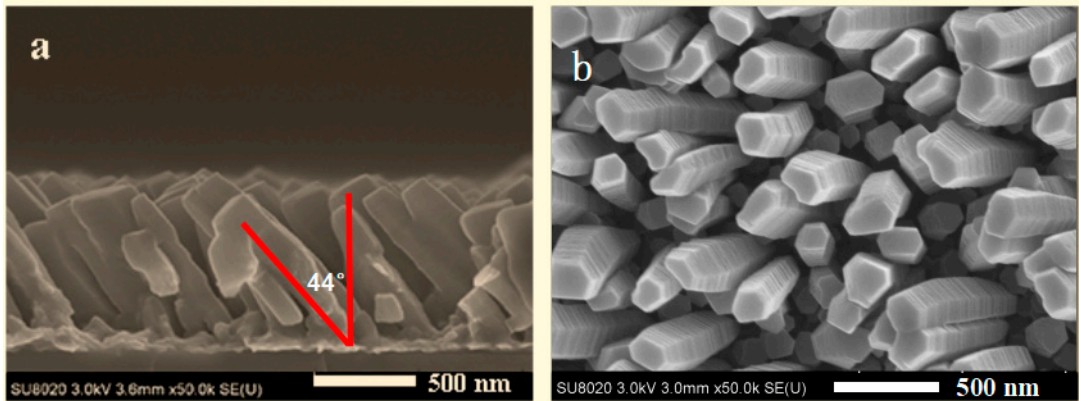

**Figure 5.** SEM (**a**) cross-section view and (**b**) surface view of a Mg nanostructured thin film synthesized for $\alpha = 85°$ and $P_{\text{Tot}} = 0.26$ Pa.

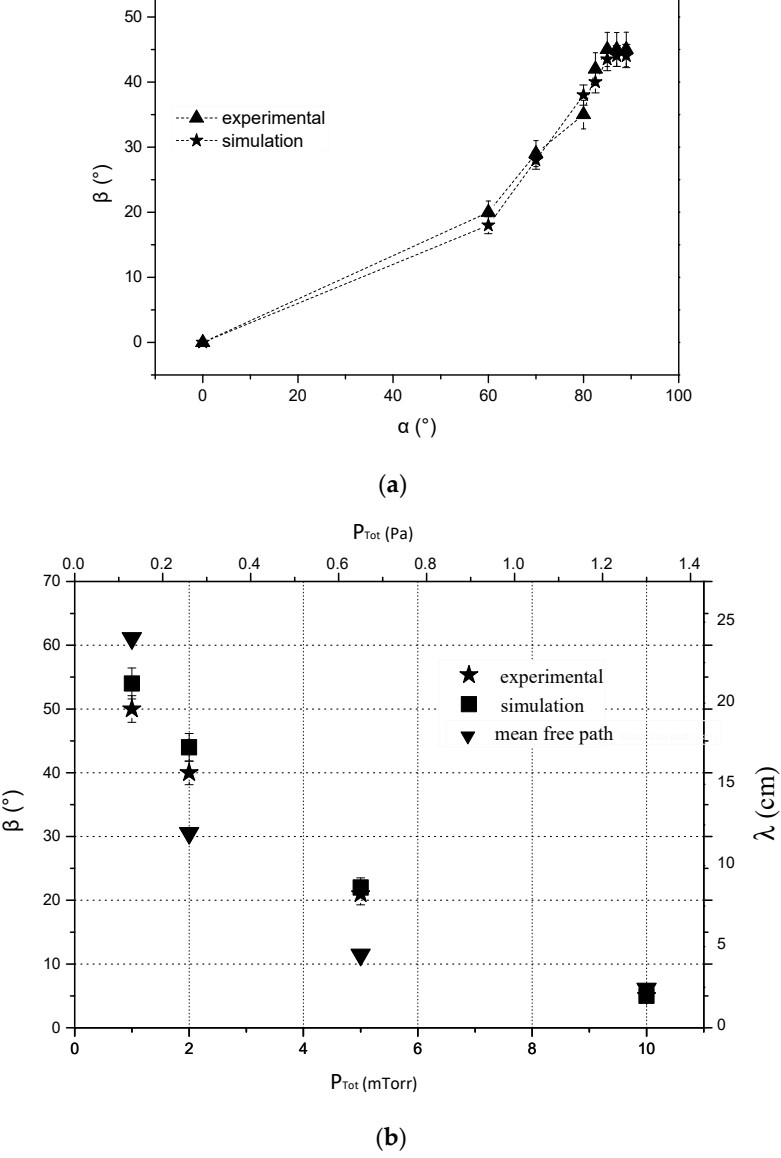

**(b)**

**Figure 6.** Evolution of the experimental and simulated values of the columnar angle β as a function of the tilt angle, $\alpha$ (**a**) and of the working pressure, $P_{\text{Tot}}$ (**b**) for Mg nano-sculpted films and the mean free path. The films were deposited for a discharge power of 50 W.

It has to be noted that since the substrate to target distance (8 cm) is not very high in our chamber, the diameter of the target (2 inches) have to be taken into account to distinguish $\alpha$ and the incident angle of the particles, since the majority of the latter comes from the racetrack region of the target. The size of the particles source induces an angle of deviation in the $\alpha$ direction which increases with the target diameter [33]. This angle of deviation has been calculated in previous work for similar conditions and slightly increases with $\alpha$ due to the geometrical inclination of the substrate leading to an asymmetric deposition. Indeed, the particles sputtered at the left side of the target have a higher probability to reach the left side of the substrate and inversely for the right side. This increases the deviation angle and can explain the similar morphologies for coatings synthesized for $\alpha > 85°$.

On Figure 6a and Supplementary Figures S1 and S2, the corresponding kMC simulations obtained by using the NASCAM (NAnoSCAle Modeling) code and the procedure described in the experimental part. As input parameters, we have utilized the defined experimental parameters (50 W, 0.26 Pa). A number of $5 \times 10^5$ atoms was chosen to obtain a film thickness similar to the one corresponding to the experimental conditions according to the size of the simulation box (X= 1000 and Y = 2 Mg atom unit). The ballistic deposition simulation of Mg atoms was used to understand the growth mechanisms of these films. The morphology of both simulated and experimental thin films were compared and the effect of the deposition parameters was analyzed. From the good agreement between the calculated and experimental data, it appears that the simulation employed in this work is perfectly adapted to our deposition.

Considering that our films are deposited without intentional heating, we can roughly estimate that the deposition temperature is about ~323 K. This corresponds to a T*, the generalized temperature of the Anders's Structure Zone Diagram (ASZM) [17], of ~0.25 since the melting temperature of Mg is 923 K. For this T* condition and considering, in our process conditions, that the normalized energy was < 1 [34], the ASZM depicts the synthesized films as a zone I film corresponding to $T_s/T_m < 0.3$, for which surface diffusion is limited, and therefore does not allow for the filling of the void regions that form in the microstructure because of the geometrical shadowing effect occurring during the GLAD experiments. In these conditions, the film growth proceeds by the formation of an underdense, fine nanofibrous microstructure that develops into a columnar morphology. In the conditions, where geometric restrictions govern the formation of the microstructure, a strong anisotropic deposition is observed. Furthermore, it has to be noted that, for the "zone I" conditions, the columnar tilt angle ($\beta = 44 \pm 1.0°$ in our case) is in line with the Tait's rule derived from geometric analysis of the inter-column shadowing geometry [34].

Figure 6b shows the evolution of $\beta$ as a function of $P_{Tot}$. In this work, $P_{Tot}$ was varied from 0.13 to 1.3 Pa where 0.13 Pa corresponds to the minimum value necessary to maintain the magnetron discharge. The sputtering power and $\alpha$ were fixed at 50 W and 85°, respectively. $\beta$ rapidly decreases as $P_{Tot}$ increases, from $\beta = 51 \pm 1.0°$ for 0.13 Pa to $\beta = 5 \pm 0.5°$ for 1.3 Pa. The modification of the columnar tilt angle can be attributed to a decrease of the collimation of the incident particle flux due to the increase of collision probability as $P_{Tot}$ increases. Indeed, this probability mainly depends on the mean free path of the sputtered Mg atoms ($\lambda_{Mg}$) which is inversely proportional to $P_{Tot}$ following this equation:

$$\lambda = \frac{\kappa_B T}{\sqrt{2}\pi d^2 P_{Tot}}$$

where, $\kappa_B$ is the Boltzmann constant ($1.380 \times 10^{-23}$ J/K), $T$ is the temperature in $K$, $P_{Tot}$ is the total pressure in Pascal, and $d$ is the diameter of the gas particles in meters. From this relation, $\lambda_{Mg}$ (atomic diameter = 1.72 Å) ranges from 24 to 2.4 cm between 0.13 and 1.33 Pa, respectively. Considering the target/substrate distance (8 cm) that is used, an increase of $P_{Tot}$ induces a large amount of collisions between particles for the 1.3 Pa conditions resulting in a less porous film. Figure 6b shows that the morphology of the simulated thin films is again in line with the experimental ones. In addition, the calculated mean free path for Ti atoms as a function of the pressure is also presented. It appears that the mean free path becomes smaller than the target-to-substrate distance for a pressure value around

0.7 Pa ($\lambda_{Mg}$ = 4.5 cm). Below this pressure, very few collisions occur through the vapor phase, whereas, a higher pressure leads to numerous collisions between particles. The analysis of the predicted film morphology at different pressures also allows determination of the range of pressure where a ballistic deposition process occurs which, in our case, is between 0.13 and 0.26 Pa, and its mean free path ranges from 24 cm to 12 cm.

If it is very difficult to measure the porosity of thin films ($\chi$) experimentally because of the low quantity of material. Therefore, it has been simulated using the PoreSTAT software [35] which uses the NASCAM files to perform a full 3D analysis of the porous structure of the material, or a 2D study on the different slices of material belonging to the XZ or YZ planes (X and Y are horizontal axes defining the substrate, whereas, Z corresponds to the vertical axe, defining the film height) [36]. Figure 7 shows the evolution of $\chi$ as a function of $\alpha$ for $P_{Tot}$ = 0.26 Pa and of $P_{Tot}$ for $\alpha$ = 85°. It appears that $\chi$ increases with $\alpha$ from 54% until 60% for $\alpha$ = 85° and then stabilizes for higher values. On the other hand, $\chi$ increases from 51% until 66% when reducing $P_{Tot}$ from 1.33 to 0.13 Pa. These evolutions are obviously correlated with the evolution of the nanosculpted films feature with the $\alpha$ and $P_{Tot}$ parameters. More precisely, it appears from Figure 8 that the evolution of $\chi$ is linearly correlated with the evolution of the aspect ratio, $\Gamma$, which is convenient since tuning the key parameters of the process such as $\alpha$ or $P_{Tot}$, we get a fine control on the porosity of the films which is indirectly correlated with the surface area of the material. This linear correlation can be understood by considering the meaning of $\Gamma$ which is the ratio of the inter columnar space on the width of columns. On the basis of this definition, it is obvious that an increase of $\Gamma$ will lead to an increase of the material porosity since there is more space between the columns because the intercolumnar space increases or the column width reduces (or both).

The chemical and structural characterization of the nano-sculpted films have again been performed by XPS and XRD measurements. As expected the XPS data reveals the presence of strong oxygen and carbon signals explained by the surface contamination which is likely even stronger for these porous films. Unfortunately, in this case, because of the nanostructured features of the material, it is not easy to depth profile the thin film. To the contrary, XRD measurements are still possible in good conditions and are reported in Figure 2 for a nano-sculpted sample synthesized for $\alpha$ = 85° and $P_{Tot}$ = 0.26 Pa (sputtering power of 50 W).

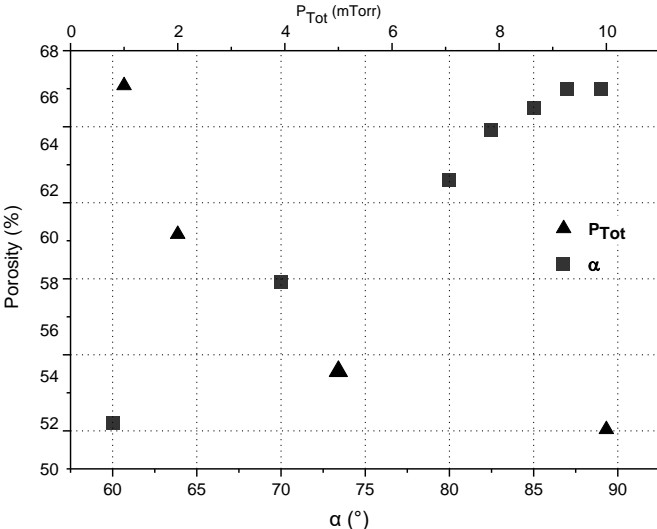

**Figure 7.** Evolution of the simulated values of the porosity as a function of the tilt angle, $\alpha$, and of the working pressure, $P_{Tot}$, for Mg nano-sculpted films prepared using a discharge power of 50 W.

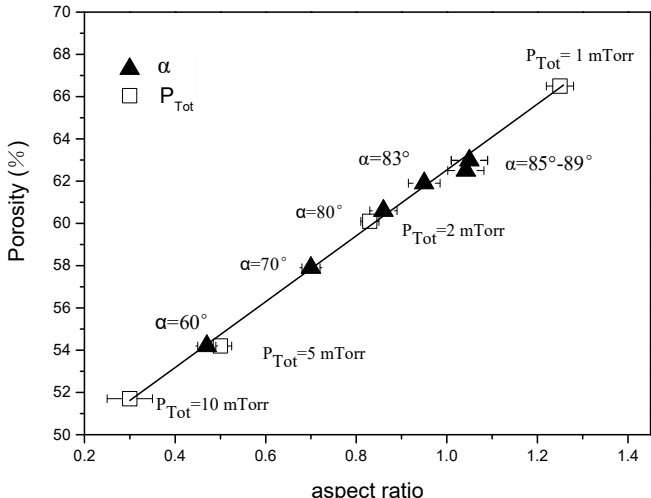

**Figure 8.** Evolution of the porosity as a function of the aspect ratio, Γ, for Mg nano-sculpted films with the $\alpha$ and $P_{\text{Tot}}$.

From the data we determine that the crystalline constitution of the material is only slightly affected by the utilization of the GLAD geometry. Indeed, all diffraction peaks observed for the dense Mg coatings are again present with the same relative intensity. The only minor difference is related to the presence of MgO lines that appear in addition to the already identified Mg lines. This suggests that the quantity of oxygen in the bulk of the material is likely higher in the nano-sculpted films allowing for the presence of MgO grains. This can be understood when considering the magnified surface area which is subjected to oxidation during the growth as compared with the situation occurring for dense film deposition.

## 4. Conclusions and Perspectives

Through the present work, we provide a fairly clear description and understanding of a magnetron sputtering in grazing angle geometry method allowing for the deposition of Mg nanocolumnar thin films for potential hydrogen storage. The effect of the deposited angle and sputtering pressure on the Mg nanocolumnar structure has been specifically investigated. The good agreement between experimental observations and model predictions indicates that the simulations realistically reproduces the competitive growth mechanism involved in GLAD experiments. On the basis of this study, we conclude that the fundamental mechanisms responsible for the growth of nano-sculpted Mg film in a MSGLAD are based on (i) the self-shadowing mechanisms at the surface and (ii) the collisional processes of the sputtered particles in the gas phase. In addition, it appears, under our experimental conditions, that the self-diffusion of deposited Mg atoms is strongly reduced and that the microstructure of our films belong to the zone I of the ASZM. In addition, we learn that when growing Mg porous films and because of the strong reactivity of Mg towards oxygen, surface and even bulk oxidation easily occurs. If not controlled, probably, the porous film would not be suitable for hydrogen storage.

**Supplementary Materials:** The following are available online at http://www.mdpi.com/2079-6412/9/6/361/s1, Figure S1: SEM cross-section view and of Mg films deposited for $P_{\text{Tot}}$ = 0.26 Pa and varying $\alpha$ from 60° to 89°. The green images correspond to the structures calculated by using Mkc modeling, Figure S2: SEM cross-section view and of Mg films deposited for $\alpha$ = 85° and varying $P_{\text{Tot}}$ from 0.13 to 1.3 Pa. The green images correspond to the structures calculated by using Mkc modeling.

**Author Contributions:** Conceptualization, H.L. and R.S.; Methodology, H.L.; Software, H.L.; validation, H.L., X.G., A.P. and D.T.; Formal analysis, H.L.; Investigation, H.L.; Resources, H.L.; Data curation, H.L.; Writing—original draft preparation, H.L.; Writing—review and editing, R.S. and W.L.; Visualization, H.L.; Supervision, R.S. and W.L.; Project administration, R.S., W.L. and M.C.; Funding acquisition, W.L. and M.C.; The design of the study, H.L., R.S.; The collection, analyses or interpretation of data, H.L., X.G., A.P. and D.T.; The writing of the manuscript, H.L.; The decision to publish the results, R.S., W.L. and M.C.

**Funding:** This research was funded by the F.R.I.A grant of the National Fund for Scientific Research (FNRS, Belgium); the Joint Foundation of National Natural Science Foundation of China, Grant No. U1764254; the National Natural Science Foundation of China, Grant No. 51871166; 321 talent projects of Nanjing (China), Grant No. 631783; 111 Project (China), D17003.

**Acknowledgments:** A.P. thanks the F.R.I.A grant of the National Fund for Scientific Research (FNRS, Belgium); H.L. thanks the Joint Foundation of National Natural Science Foundation of China (Grant No. U1764254); the National Natural Science Foundation of China (Grant No. 51871166); 321 talent projects of Nanjing (Grant No. 631783), China; 111 Project (D17003), China.

**Conflicts of Interest:** The authors declare no conflict of interest.

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
