# Peer review of "Experimental and Modeling Study of the Fabrication of Mg Nano-Sculpted Films by Magnetron Sputtering Combined with Glancing Angle Deposition"

_coatings, doi:10.3390/coatings9060361_

Reviewer 1 Report

In this work, the authors describe the fabrication of Magnesium nano-sculpted vfilms by using magnetron sputtering in a glancing angle deposition. The prepared films are thought to be adequate for the  storage of hydrogen. The manuscript is well written, the procedure is clearly explained and the methods of characterization are well selected. Some minor corrections are  needed:

Title: I think it would be better to use in the title and abstract Magnesium and "fabrication"  instead of synthesis as it is a physical method of deposition.

line 22: highlighted

' 24: development of an air stable material...

' 34: one of the promising ones

'35 compared to..

'48: low density

'61 synthesize (deposited, fabricated is better as this is a physical process)

line 106: The cheical compostion

197: of a polycrystalline material

line 296: Therefore, it has been simulated..

line 318: to the contrary

Figure 8 has to be discussed more in detail, it is not clear why films with higher aspect ratios are more porous.

line 327: surface area which is subjected to..

line 338:  under our experimental conditions

line 342: it should be stressed that porous films will, probably, not be suitable for hydrogen storage

Author Response

We would like to thank the reviewers for their detailed checking and helpful comments for our manuscript. The major improvement has been carried out in the revised manuscript.

1. I think it would be better to use in the title and abstract Magnesium and "fabrication"  instead of synthesis as it is a physical method of deposition. 

Thanks for the advice. The word of synthesis is replaced by fabrication in the title and abstract.

2. Figure 8 has to be discussed more in detail, it is not clear why films with higher aspect ratios are more porous.

Thanks for the advice. The discussion of Figure 8 has been included in the revised manuscript, and is shown in yellow color. The sentences  This linear correlation can be understood by considering the meaning of Γ which is the ratio of the inter columnar space on the width of columns. Based on this definition, it is obvious that an increase of Γ will lead to an increase of the material porosity since the is more space in between the columns because the intercolumnar space increases or the column width reduces (or both).” have been included in the revised manuscript at line 317-323, P10.

3. line 22: highlighted

Thanks for the advice. The word of understood is replaced by highlighted at line 23.

4. ‘24: development of an air stable material

Thanks for the advice. The word of an was added at line 25.

5. ' 34: one of the promising ones

Thanks for the advice. The words of one of the promising one have been revised to one of the promising ones.

6. "35 compared to

Thanks for the advice. The word of compare to have been replaced by compared to.

7. " 48: low density

Thanks for the advice. The word of light have been revised to by low at line 50.

8. "61 synthesize (deposited, fabricated is better as this is a physical process)

Thanks for the advice. The word of synthesize has been revised to fabricate at line 63-64.

9. line 106: The cheical compostion

Thanks for the advice. The words of chemical composition have been replaced by the chemical composition at line110.

10. 197: of a polycrystalline material

Thanks for the advice. The words of on a polycrystalline material have been revised to of  a polycrystalline material at line 203.

11. line 296: Therefore, it has been simulate

Thanks for the advice. The word of “Therefore” was added at line 309.

12. line 318: to the contrary

Thanks for the advice. The word of at has been replaced by to at line 334.

13. line 327: surface area which is subjected to

Thanks for the advice. The words of “surface area that is presented of” have been revised to “surface area which is subjected to” at line 343.

14. line 338:  under our experimental conditions

Thanks for the advice. The word of that was revised to under at line 355.

15. line 342: it should be stressed that porous films will, probably, not be suitable for hydrogen storage

Thanks for the advice. The sentence of “this could be detrimental for some application such as the utilization of these porous Mg coatings for hydrogen storage” has been revised to the porous film wont be suitable for hydrogen storage at line 354-355.

Reviewer 2 Report

The authors have described an improved technique of using nano-scuplted Mg flims that could be a promising material for hydrogen storage.

They have carried out fundamental scientific experiments and characterization work and co-related with simulation model. They have initiated studies and concluded that oxidation still occurs on growing Mg depositions. One of the reasons of this study was to reduce the oxidation by downsizing the Mg nanoparticles below 100nm. Therefore, further studies are required to be carried out or perhaps the team would need to look at alternative materials for hydrogen storage.

Overall, the paper is well written with adequate and reasonable arguments. There are some minor queries:

On Page 2 LN 49, Mg based hydrides have reversibility in comparison with other hydrides. Could the authors be more specific on the mode of reversibility?

Page 2 LN 72, the phrase can be controlled can be grown needs editing. I am not sure which of the words are to be omitted or is there a missing word to complete the sentence.

Author Response

We would like to thank the reviewers for their detailed checking and helpful comments for our manuscript. The major improvement has been carried out in the revised manuscript.

1.On Page 2 LN 49, Mg based hydrides have reversibility in comparison with other hydrides. Could the authors be more specific on the mode of reversibility?

The H2 storage system absorb H2 and also release H2 under certain temperature/pressure, performing the reversibility absorption/desorption cycles for several times, and keep stable absorption/desorption performance. Mg based hydrides can be considered as an efficient hydrogen storage system due to the abundance, low cost, low density, low toxicity, high hydrogen capacity and high efficiency sorption property. Mg based hydrides included Mg-based complex (Mg(AlH4)2, LiMg(AlH4)3, and Mg(BH4)2) and elemental hydrides (MgH2) system. Refer to the reported results, the hydrogen desorption reaction for Mg-based complex system usually has multistep reaction, which would limit the hydrogen absorption reaction of the system. Mg based elemental hydrides system (Mg/MgH2) has much potential for application, the H2 absorption/desorption chemical reaction are related to sorption temperature/pressure by adjusting the size of Mg and using the catalyst. We would focus on developing the nanostructured Mg/MgH2 system to store H2.

2. Page 2 LN 72, the phrase can be controlled can be grown needs editing. I am not sure which of the words are to be omitted or is there a missing word to complete the sentence.

Thanks for the advice. The words of  can be grown” have been deleted in the revised manuscript.

Reviewer 3 Report

The Authors reports some interesting experimental procedures on the deposition of Mg thin films. Overall the paper is well written and the conclusions are adequate. A minor revision: Figure 1 is very blurry, please replot.

Author Response

We would like to thank the reviewers for their detailed checking and helpful comments for our manuscript. The major improvement has been carried out in the revised manuscript.

Response to Reviewer 3 Comments

1. The Authors reports some interesting experimental procedures on the deposition of Mg thin films. Overall the paper is well written and the conclusions are adequate. A minor revision: Figure 1 is very blurry, please replot.

Thanks for the advice. A clear picture has been added in Figure 1 at line 93 and marked by yellow color.